# Petrogenesis of Low Sr and High Yb A-Type Granitoids in the Xianghualing Sn Polymetallic Deposit, South China: Constrains from Geochronology and Sr–Nd–Pb–Hf Isotopes

**ChangHao Xiao *** **, YuKe Shen and ChangShan Wei**

Laboratory of Dynamic Diagenesis and Metallogenesis, Institute of Geomechanics,
Chinese Academy of Geological Sciences, Beijing 100081, China; shenyuke@geomech.ac.cn (Y.K.S.);
weichangshan@geomech.an.cn (C.S.W.)
* Correspondence: xiaochanghao1986@126.com or xiaochanghao@geomech.ac.cn; Tel.: +86-010-88815612

**Abstract:** The nature and origin of the early Yanshanian granitoids, widespread in the South China Block, shed light on their geodynamic setting; however, understanding their magmatism processes remains a challenge. In this paper, we present both major and trace elements of bulk rock, Sr–Nd–Pb isotopic geochemistry, and zircon U–Pb–Hf isotopes of the low Sr and high Yb $A_2$-type granites, which were investigated with the aim to further constrain their petrogenesis and tectonic implications. Zircon U–Pb dating indicates that these granites were emplaced at ca. 153 Ma. The granites are characterized by high $SiO_2$ (>74 wt.%) and low $Al_2O_3$ content (11.0 wt.%–12.7 wt.%; <13.9 wt.%). They are enriched in large ion lithophile elements (LILEs) (e.g., Rb, Th, U, and K) and Yb, but depleted in high field-strength elements (HFSEs) (e.g., Nb, Ta, Zr and Hf), Sr, Ba P, Ti and Eu concentrations. They exhibit enriched rare earth elements (REEs) with pronounced negative Eu anomalies. They have $\varepsilon_{Nd}(t)$ values in a range from −6.5 to −9.3, and a corresponding $T_{DM}$ model age of 1.5 to 1.7 Ga. They have a $(^{206}Pb/^{204}Pb)_t$ value ranging from 18.523 to 18.654, a $(^{207}Pb/^{204}Pb)_t$ value varying from 15.762 to 15.797, and a $(^{208}Pb/^{204}Pb)_t$ value ranging from 39.101 to 39.272. The yield $\varepsilon_{Hf}(t)$ ranges from −6.1 to −2.1, with crustal model ages ($T_{DMC}$) of 1.3 to 1.6 Ga. These features indicate that the low Sr and high Yb weakly peraluminous $A_2$-type granites were generated by overlying partial melting caused by the upwelling of the asthenosphere in an extensional tectonic setting. The rollback of the Paleo-Pacific Plate is the most plausible combined mechanism for the petrogenesis of $A_2$-type granites, which contributed to the Sn–W polymetallic mineralization along the Shi-Hang zone in South China.

**Keywords:** zircon U–Pb geochronology; Sr–Nd–Pb–Hf isotopes; low Sr and high Yb $A_2$-type granite; Xianghualing; South China

## 1. Introduction

The Mesozoic Age marks an important period in the geologic evolution of mainland South East Asia (Figure 1a) [1–4], during which extensive magmatism took place in the South China Block (SCB) parallel to the present-day coastline (Figure 1b). This magmatic zone was preliminarily named the "Shi-Hang zone" after Gilder et al. [4]. As a granitoid belt, it reveals the Nd-depleted mantle model ages ($T_{DM}$ = 1.4 ± 0.3 Ga) and negative $\varepsilon_{Nd}(t)$ values (−4 to −8) in composition [5]. Such Sn–W polymetallic metallogenic domains related to widespread granitoids in South China represents one of the largest tungsten-tin polymetallic ore provinces on Earth [6–15]. Previous studies demonstrated that these temporally and spatially associated intrusions are different in terms of petrography, and elemental and

isotopic compositions, and argued for different petrogenesis and tectonic regimes, ranging from an extension [11], rift-related and convergent [5,16–19], or subduction-related models (Figure 1d) [20–24].

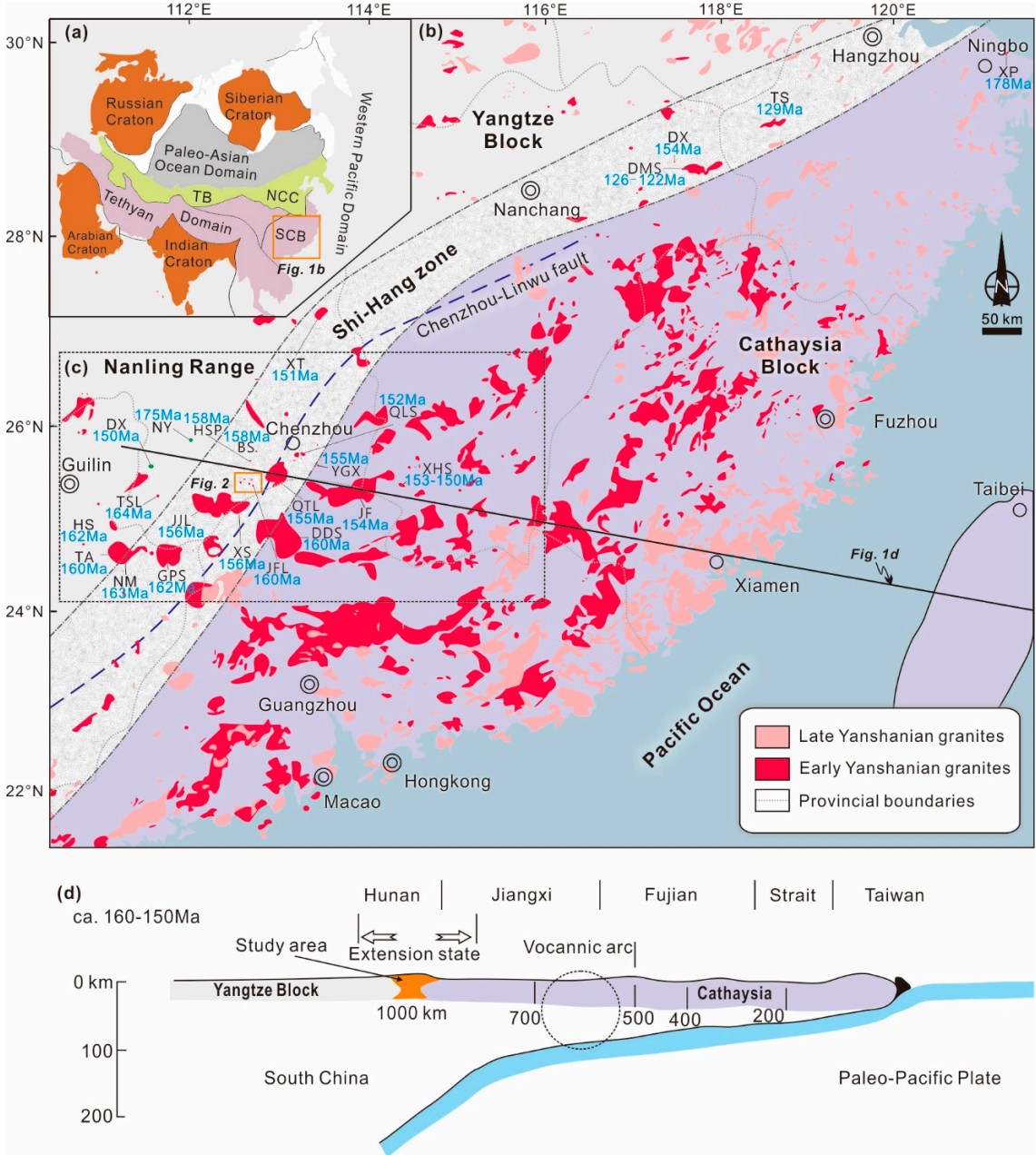

**Figure 1.** (**a**) Simplified tectonic map of Asia showing the framework of the joint area of the Paleo–Asian, Tethyan, and Pacific domains [4]. TB, Tibet Block; NCC, North China Block; SCB, South China Block. (**b**) Distribution of the Mesozoic intrusive rocks in South China [7,15,21,24,25]. The locations of the Shi-Hang zone and Nangling range are after [19,26,27]. (**c**) Sketch map of the Nanling Range [13]. (**d**) Simplified map showing the Late Jurassic subduction in Southeast China [28]. NM, Niumiao granodiorite [29]; TA, Tong'an monzonite [29]; HS, Huashan granite [30,31]; GPS, Guposhan granite [30,31]; TSL, Tongshanling granodiorite [19,32]; JJL, Jinjiling granite [19,33]; XS, Xishan granite [33]; DDS, Dadongshan granite [34]. JFL, Jianfengling (our no published data); DX, Daoxian basalt [19,35]; NY, Ningyuan basalt [19,35]; QTL, Qitianling granite [36]; HSP, Huangshaping granite [37]; BS, Baoshan granodiorite [32,38]; JF, Jiufeng granite [39,40]; QLS, Qianlishan granite [41]; YGX, Yaogangxian granite [42]; XHS, Xihuashan granite [43,44]; XT, Xitian granite [15]; DMS, Damaoshan granite [45]; DX, Dexing granodiorite [46]; TS, Tongshan granite [45]; XP, Xiepu granite [24].

The Xianghualing Sn polymetallic deposit is situated at the boundary between the Paleo-Pacific and the Tethyan tectonic domains (Figure 1a). The Late Jurassic granites related to this deposit could provide constraints on the petrogenesis of Late Jurassic granite-related W–Sn mineralization and their tectonic settings. In this contribution, we present major and trace element geochemistry, Sr–Nd–Pb isotope data for these granites, and zircon U–Pb dating and Hf isotopic evidence.

## 2. Geological Background

The Yangtze Block to the northwest and the Cathaysia Block to the southeast were amalgamated to form the South China Block during the Neoproterozoic Era (Figure 1b) [7,11]. Numerous granites are distributed in the Cathaysia Block and the Shi-Hong zone. The study area is located in the central part of Shi-Hang zone (Figure 1b). The NE–SW trending of the Chenzhou-Lingwu fault zone represents a regional fault in the study area (Figure 1b) [25,47]. The zone was originally formed at 970 Ma [1] and reactivated during the Triassic and the Cretaceous Eras [25]. A number of granite plutons and associated Sn polymetallic mineralization were distributed along the Chenzhou-Lingwu fault belt to the west of Shi-Hong zone (Figure 1b). This Sn-metallogenic region, called the Nanling Range, is characterized by multiple and diverse mineral deposits (W, Sn, Cu, Pb–Zn, etc.) and the Jurassic-Cretaceous intrusions [13]. The granites include the Baoshan, Tongshanling, Niumiao, Yuanzhuding, Guposhan, Huashan, Xishan, Jinjiling, Qianlishan, and Qitianling plutons (Figure 1b). Ore deposits associated to this belt are: Shizhuyuan (W–Sn–Mo–Bi–F) [48], Furong (Sn) [6,49,50], Yaogangxian (W) [49], Xianghualing (Sn–W–Pb–Zn) [12,51], Huangshaping (Pb–Zn–W–Sn) [7,52], Baoshan (Cu–Pb–Zn–W) [38], and Yuanzhuding (Cu–Mo) [14]. Basaltic rocks also crop out near the Chenzhou-Lingwu fault, such as the Daoxian and Ningyuan basalts (Figure 1b) [19,35].

The Xianghualing Sn-polymetallic deposit is situated at the Midwestern point of the Nanling Range (Figure 1c). The Xinfeng mine is one of the most important mines in the Xianghualing deposit. The Xianghualing area is a tectono-magmatic dome (Figure 2a). The exposed basement comprises the Cambrian slate and metasandstone, sandstone and shale in the Middle Devonian Tiaomajian Formation, limestone and dolomitic limestone in the Middle Devonian Qiziqiao Formation, dolomitic limestone and sandstone in the Upper Devonian of the Shetianqiao Formation, and the Carboniferous carbonate and clastic rocks. The Qiziqiao Formation is the major ore bed, and the Laiziling and Jianfengling are the two largest granites which intruded into the Cambrian and Devonian Eras, respectively. The Xianghualing Sn polymetallic deposit is a typical skarn deposit related Laiziling Pluton [12]. Skarn-type ore bodies and vein-like bodies formed in the contact zone of the granites.

## 3. Sampling and Analytical Methods

The granite samples in this study included ZK65-1, ZK65-2, ZK65-3 and ZK57-1 and were collected from drill cores ZK65 and ZK57, one of the deepest holes of Laiziling pluton prior to 2015, in the Xinfeng mine, Xianghualing deposit (Figure 2b). The sequence from the granite to the country rocks is as follows: granite → altered sandstone → skarn → Sn polymetallic ores in the drill cores ZK65, or, alternatively, granite → skarn → tungsten–bearing quartz veins → ore-bearing marble → fracture in drill core ZK57 (Figure 2b). The pluton in the mine connects with the Laiziling pluton that exposes on the surface. Three granite samples collected from the drill core ZK65 are mainly composed of fine to medium grained porphyritic biotite monzonitic granites with a porphyritic texture. They contain 30–35% rounded, variably resorbed quartz phenocrysts of 1–8 mm in diameter, 10–15% euhedral-subhedral K-feldspar phenocrysts of 5–10 mm in length, about 5–10% equant and tabular plagioclase phenocrysts of 0.5–2 mm in length, biotite (3–8%), and minor to trace amounts of hornblende. (Figure 3a,b). Some K-feldspar phenocrysts have partially experienced sericitization. Sample ZK57-1 is two-mica monzonitic granite with small phenocrysts and finer-grained quartz (30–35%), K-feldspar (10–15%), plagioclase (10–15%), biotite (3–8%), and muscovite (3–8%) (Figure 3c,d). Accessory minerals are composed of zircon with minor apatite.

Details of analytical methods, namely whole-rock major and trace element compositions, Sr–Nd–Pb isotopes, and zircon U–Pb and Lu–Hf isotope analyses are presented in the Supplementary material.

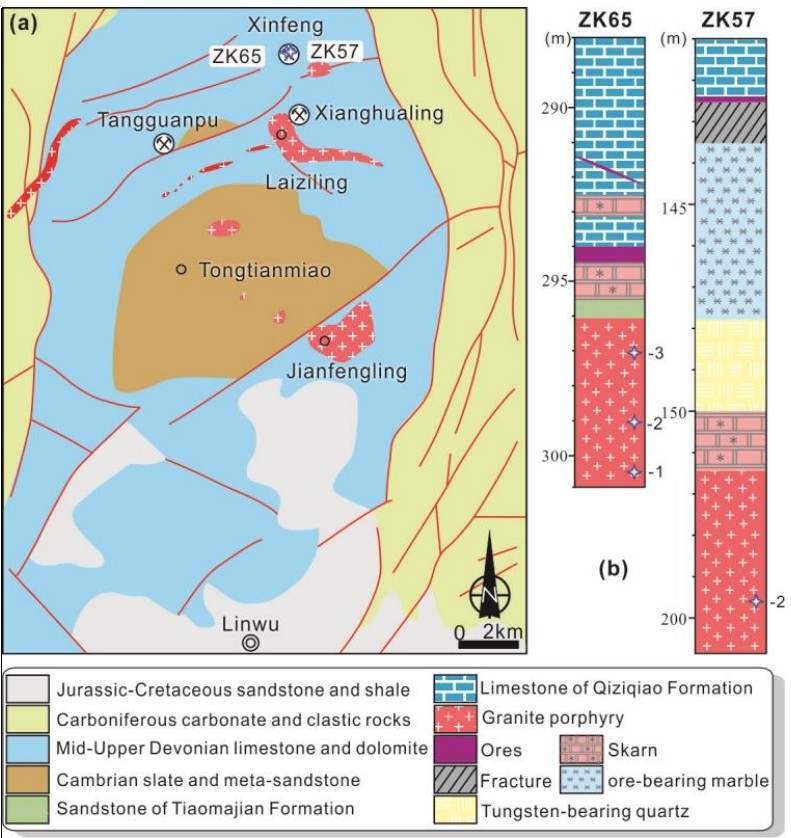

**Figure 2.** (**a**) The simplified geological map of the Xianghualing Sn-polymetallic deposit, South China [12,53]. (**b**) Drill core profile showing the sample locations.

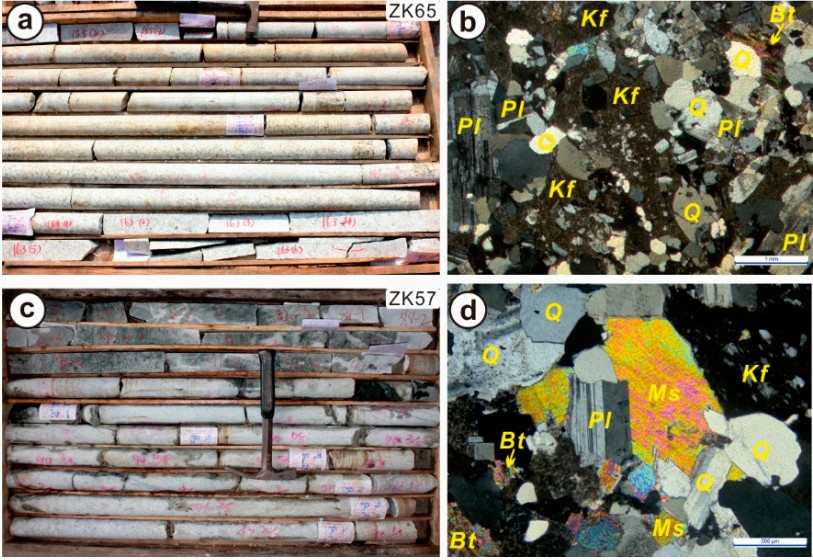

**Figure 3.** Drill cores and micrographics of ZK65 and ZK57 from the Xinfeng mine, Xianghualing area, South China. (**a**,**b**) Biotite monzonitic granites from ZK65. (**c**,**d**) Two-mica monzonitic granite from ZK57. Q, quartz; Kf, feldspar; Bt, biotite; Ms, muscovite; Pl, plagioclase.

## 4. Results

### 4.1. Whole-Rock Geochemistry

#### 4.1.1. Major and Trace Elements

The analytical results of the four granite samples are presented in Table S1. These granites have characteristics of high $SiO_2$ content (74.1 wt.%–78.0 wt.%) (Figure 4a), similar to that of the Jianfengling granites (73.6 wt.%–75.2 wt.%) [53], and Qitianling granites (65.9 wt.%–75.7 wt.%) in this region [54,55]. They are enriched in alkalis, with the $K_2O + Na_2O$ contents ranging from 7.2 wt.% to 9.2 wt.% (average 8.3 wt.%), and $K_2O/Na_2O$ ratios higher than 1. The samples are plotted in the high K calc-alkaline field in the $Na_2O + K_2O - CaO$ vs. $SiO_2$ diagram (Figure 4a). They exhibit low $Al_2O_3$ contents (11.0 wt.% to 12.7 wt.%), and A/CNK ratios (1.0–1.1) lower than 1.1, indicating their weak peraluminous affinity in the A/NK vs. A/CNK diagram (Figure 4b). They have low $Fe_2O_3$ (0.1 wt.%–0.5 wt.%), MgO (0.1 wt.%), CaO (0.6 wt.%–0.8 wt.%), $TiO_2$ (0.1 wt.%), and $P_2O_5$ contents.

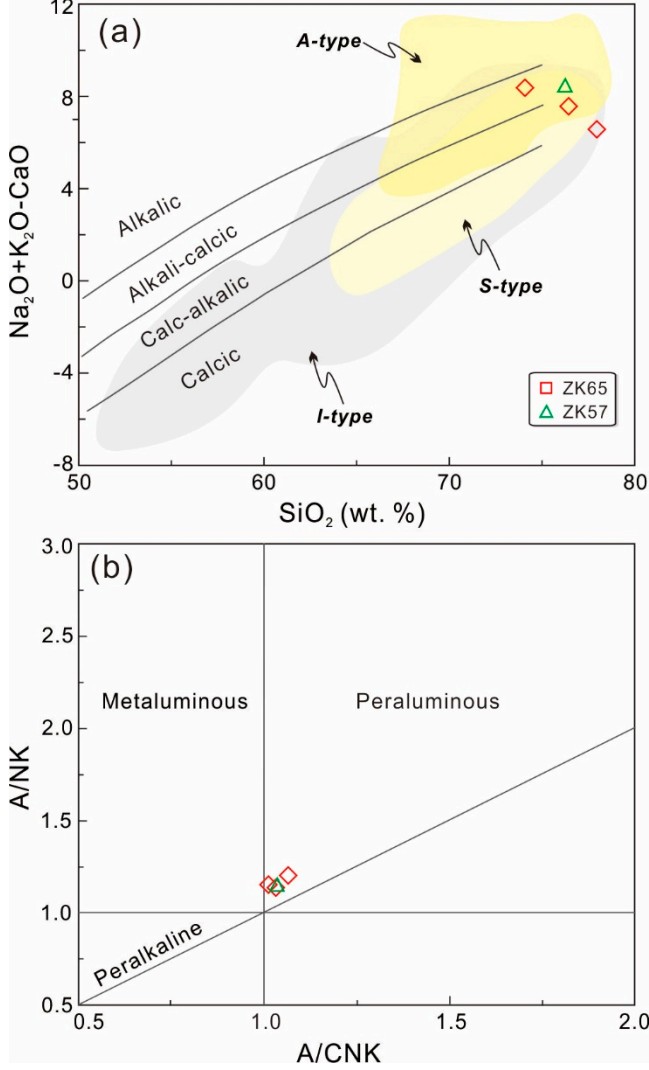

**Figure 4.** (**a**) Diagram of $SiO_2$ vs. $Na_2O + K_2O - CaO$ [56]; (**b**) Diagram of Al/(K+Na) vs. Al/(Ca + K + Na) [57].

A primitive mantle-normalized trace element diagram and chondrite-normalized rare earth element (REE) patterns for the four granite samples are illustrated in Figure 5a,b, respectively. All samples have low Sr (3.6–7.3 ppm) and Nb (0.3–2.5 ppm, except ZK65–1) contents and extremely high Rb (1606–2351 ppm) and Yb (7.9–14 ppm) contents. They are enriched in large-ion lithophile

elements (LILEs) (e.g., Rb, Th, U, and K) and depleted in high field strength elements (HFSEs) (e.g., Nb, Ta, Zr and Hf), Sr, and Ba (Figure 5a). The total contents of Zr, Nb, Ce, Y of four samples range from 430 to 646 ppm (Figure 6). The granites exhibit enriched REEs (except Eu) with a total REE ranging from 233 to 312 ppm. They are characterized by a slight enrichment of light REE (LREE) ($(La/Yb)_N$ = 1.8–5.0) and flat heavy REE (HREE) with pronounced negative Eu anomalies ($\delta$Eu = 0.02) (Figure 5b). The differentiation indexes (DI) of these samples are 93.7–94.58 (Table S1).

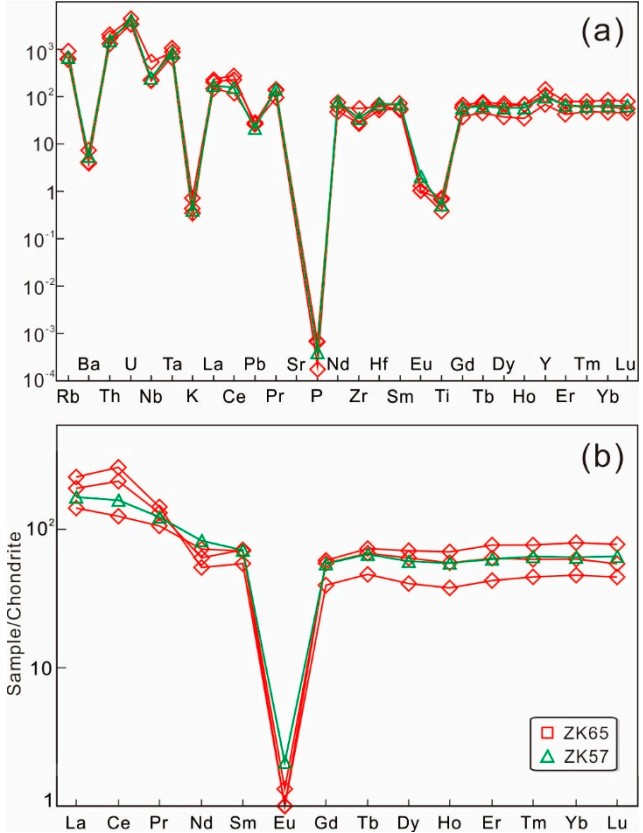

**Figure 5.** (**a**) Primitive mantle-normalized trace element spider diagram; (**b**) Chondrite-normalized REE patterns for the granites from the Xinfeng mine, Xianghualing area, South China. Normalized values for primitive mantle and chondrite are from Sun and McDonough [58].

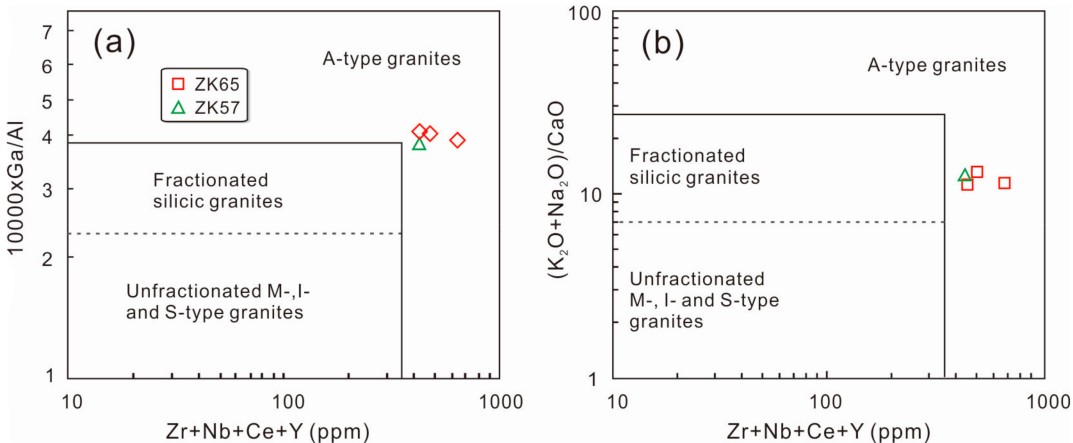

**Figure 6.** (**a**) 10,000 × Ga/Al and; (**b**) ($K_2O$ + $Na_2O$)/CaO vs. Zr + Nb + Ce + Y (ppm) diagrams showing that the Jurassic low Sr and high Yb granites are A-type granites [59].

### 4.1.2. Sr–Nd–Pb Isotopic Compositions

The Sr–Nd–Pb isotopic composition data for the granites are presented in Table S2. Initial values of $(^{206}Pb/^{204}Pb)_t$, $(^{207}Pb/^{204}Pb)_t$, and $(^{208}Pb/^{204}Pb)_t$ are calculated using zircon ages of 153 Ma.

The granites have an extremely high $^{87}Sr/^{86}Sr$ ratio (up to 1.89), which may be due to their extreme Sr (also Eu and Ba) depletion, and thus a very high Rb/Sr ratio. Previous studies pointed that the initial of Sr isotope is unreliable of high Rb granite [60]. We therefore do not discuss the initial ratio of Sr isotopes on the petrogenesis of the Xianghualiang granites. They have $\varepsilon_{Nd}(t)$ values ranging from $-9.3$ to $-6.5$ and $(^{143}Nd/^{144}Nd)_i$ from 0.511963 to 0.512107, corresponding to depleted mantle model ages ($T_{DM2}$–Nd) of 1471–1702 Ma. The plots of $\varepsilon_{Nd}(t)$ vs. $^{206}Pb/^{238}U$ age is exhibited in Figure 7a.

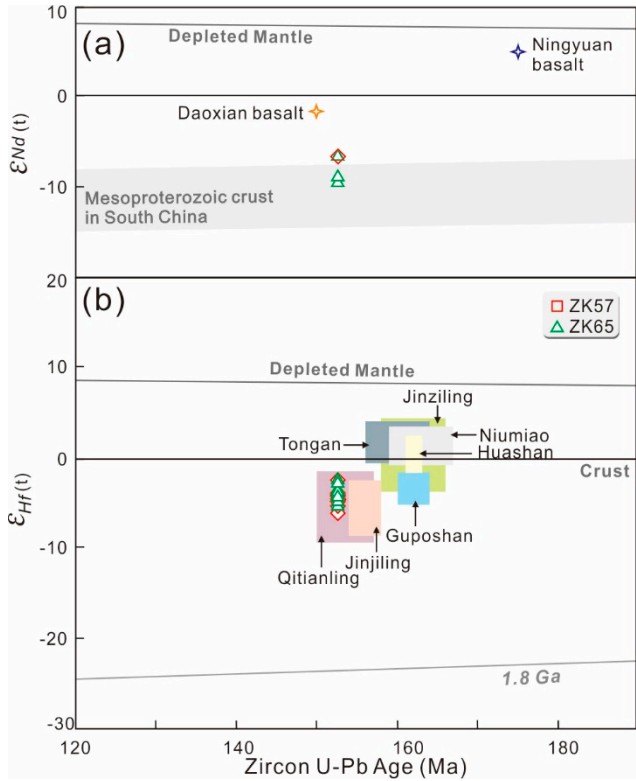

**Figure 7.** (**a**) $\varepsilon_{Nd}(t)$-age diagram and (**b**) $\varepsilon_{Hf}(t)$-age diagram for zircons from Xianghualing area. The data of zircon U–Pb age, $\varepsilon_{Nd}(t)$ and $\varepsilon_{Hf}(t)$ are from [7,15,21,24–45].

Four samples are characterized by high radiogenic Pb isotopic composition, with present-day rock $^{206}Pb/^{204}Pb$, $^{207}Pb/^{204}Pb$, and $^{208}Pb/^{204}Pb$ ratios of 19.624 to 19.791, 15.816 to 15.847, and 39.478 to 39.795, and their corresponding initial ratios are 18.523 to 18.654, 15.762 to 15.797, and 39.101 to 39.272, respectively.

### 4.2. Zircon U–Pb Ages

Most zircon grains separated from four granite samples are euhedral and prismatic with aspect ratios of 1:1–1:2 and lengths of 50–150 μm. They are transparent and light yellow under an optical microscope. Ubiquitous simple internal oscillatory zoning and little inherited cores are observed by Cathodoluminescence (CL) images (Figure 8). Such characteristics indicate they are magmatic zircon in origin [61]. A few zircon crystals exhibit complex secular zonings (Figure 8).

LA-ICP-MS zircon U–Pb isotopic data for these granitoids are shown in Table S3. The zircon U–Pb concordia and weighted mean diagrams are illustrated in Figure 9. These zircons have varied U (154–6771 ppm) and Th (70–4505 ppm) contents, with Th/U ratios ranging from 0.3 to 0.8 (Table S3), indicating that they are magmatic in origin [62]. Thirteen analyses fall on the Concordia in a single

group from the sample ZK57, yielding a Concordia age of 152.8 $\pm$ 0.6 Ma (MSWD = 0.2, *n* = 13). The remaining three analyses have a $^{206}$Pb/$^{238}$U age of 148.3 $\pm$ 1.5 Ma and 147.1 $\pm$ 1.8 Ma (Figure 9a). Fifteen spot analyses of zircons from sample ZK65 yield a single $^{206}$Pb/$^{238}$U age population of 105–156 Ma with Concordia age of 152.7 $\pm$ 2.0 Ma (MSWD = 2.9, *n* = 9) (Figure 9b and Table S3). The two Concordia ages may represent the crystallization age of the granites.

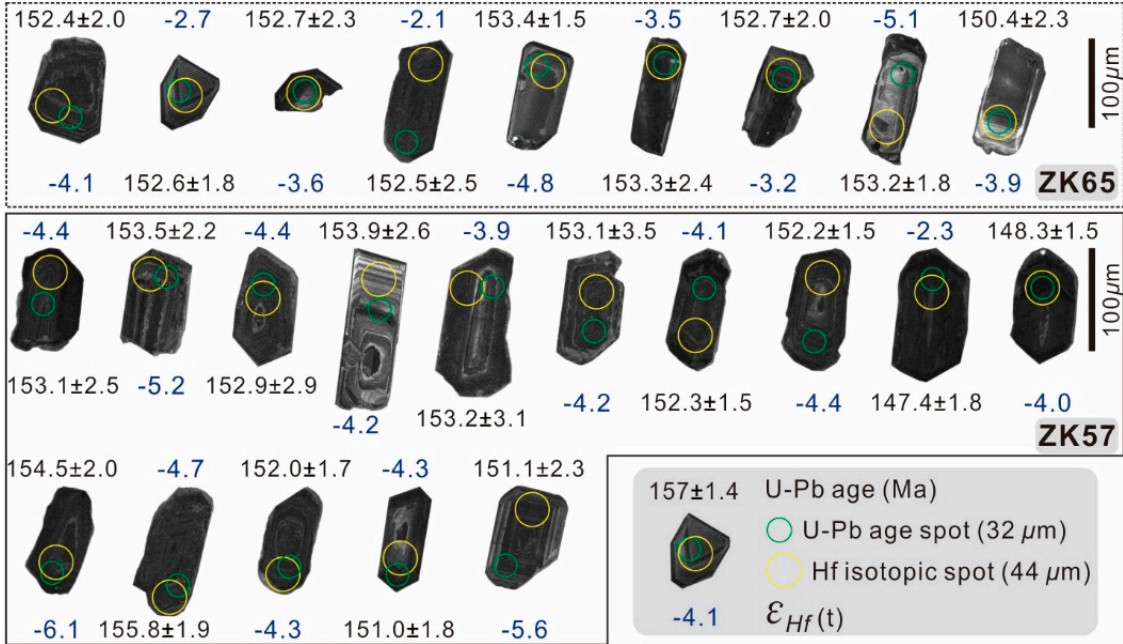

**Figure 8.** CL electron micrographs of representative zircons separated from the drill cores of ZK65 and ZK57 from the Xinfeng mine, Xianghualing area, South China. Spots are tested by in situ U–Pb age and Hf isotope.

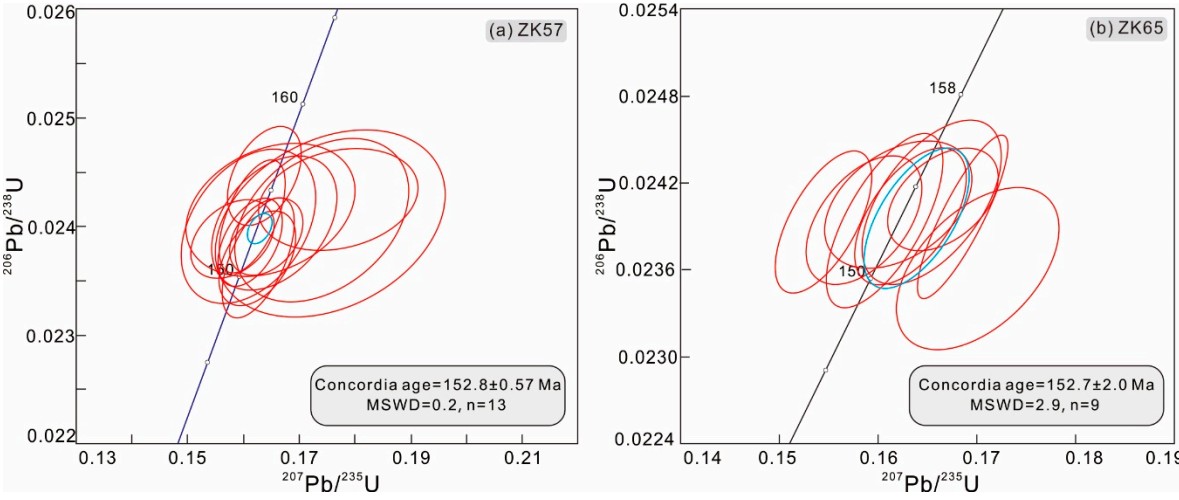

**Figure 9.** LA–CP–MS zircon U–Pb Concordia diagrams for the drill cores of ZK57 (**a**) and ZK65 (**b**) from the Xinfeng mine, Xianghualing area, South China.

### 4.3. Zircon Hf Isotopes

The Lu–Hf isotopic compositions of zircon crystals separated from the two granite samples from the Xianghualing area in South China are listed in Table S4. The plots of $\varepsilon_{Hf}$(t) vs. $^{206}$Pb/$^{238}$U age is illustrated in Figure 7b.

The zircons have $^{176}Lu/^{177}Hf$ ratios from 0.000776 to 0.003445 and $^{176}Hf/^{177}Hf$ ratios from 0.282514 to 0.282628 (Table S4). Most of $^{176}Lu/^{177}Hf$ ratios of zircon from ZK57 are below 0.002, and therefore the accumulation of radiogenic Hf after the formation of zircons can be ignored. $^{176}Lu/^{177}Hf$ ratios of zircon from ZK65 are a little large than 0.002 except for two analyses (ZK65-1, ZK65-5), indicating a minor radiogenic production of $^{176}Hf$. The $\varepsilon_{Hf}(t)$ values for analyses of zircon from the samples ZK57 and ZK65 vary from $-6.1$ to $-2.1$ (Figure 7b) and yield crustal model ages ($T_{DMC}$) for the zircon crystals from the samples that have a spectrum from 1336 Ma to 1588 Ma. The $\varepsilon_{Hf}(t)$ values of ZK57 are close and smaller than those of ZK56 (Table S4).

## 5. Discussion

### 5.1. Origin of Granitic Rocks: An $A_2$-Type Affinity

The later Jurassic granitic rocks from the Xianghualing area exhibit features of high silica (74.1–78.0 wt.%) and weak peraluminosity (A/CNK ratios = 1.0–1.1). They have high $K_2O$ contents ($K_2O/Na_2O$ ratios = 1.4–4.3) and low CaO, Ba, and Sr contents. They are enriched in Nb, and Ta, and HFSE contents (Zr + Nb + Ce + Y = 430–646 ppm) (Figure 6b). They have HFSE contents higher than those of A-type granites (350 ppm) [59]. Chondrite-normalized REE plots show relatively flat patterns with large negative Eu anomalies. These features are consistent with those of A-type granites rather than S and I-types [63–66]. They are characterized by extremely low $P_2O_5$ abundances and limited phosphate minerals, compared with S-type granites.

Strontium contents have been acknowledged as a discriminating parameter to classify granites of A-type granites [63]. The granitic rocks from the Xianghualing area have extremely lower Sr contents (<10 ppm) than typical calc-alkaline I-type granites. Zhang et al. demonstrated that A-type granites in South China are characterized by low Sr and high Yb, using a Sr–Yb diagram with obvious V-type REE patterns [67]. The much lower Sr contents therefore suggest that the granites from the Xianghualing area belong to A-type granites. Biotite and hornblendes commonly exist as interstitial clots or grains (Figure 10a). Moreover, micrographic intergrowths of quartz and myrmekite usually develop in or around the alkali feldspar (Figure 10). These features are consistent with those of typical A-type granites [59,68]. In the 10,000×Ga/Al and ($K_2O + Na_2O$)/CaO vs. Zr + Nb + Ce + Y (ppm) diagrams, as well as the $SiO_2$ vs. $Na_2O + K_2O - CaO$ diagrams, they plot within A-type granites field (Figures 4a and 6a,b). All the samples fall into the $A_2$-type granites field in the plot of Ce/Nb versus Y/Nb and Nb-Y-3Ga discrimination diagrams (Figure 11a,b), and this is also evident in the plot of the R1-R2 diagram [69] (Figure 11c).

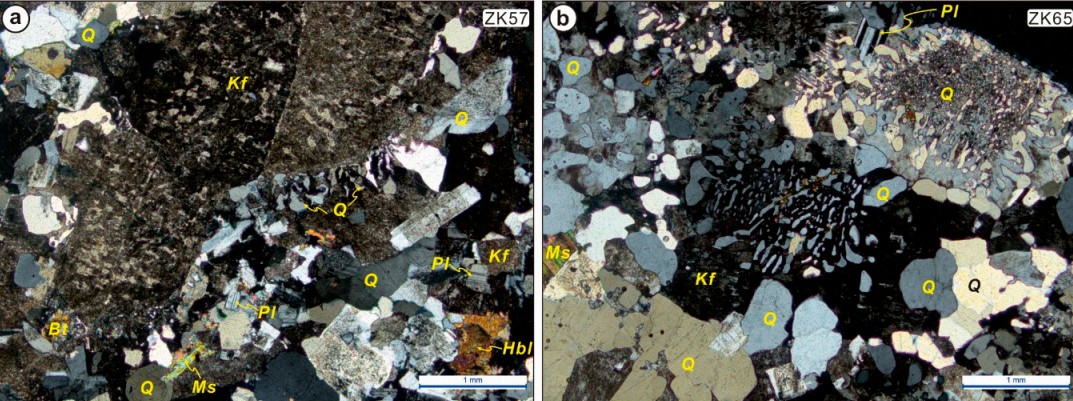

**Figure 10.** (**a**) Micrographic intergrowths of quartz and myrmekite develop around the alkali feldspar from the ZK57; (**b**) The alkali feldspar was replaced by the quartz from the ZK65. Q–quartz, Kf–feldspar, Bt–biotite, Ms–muscovite, Pl–plagioclase; Hbl–hornblende.

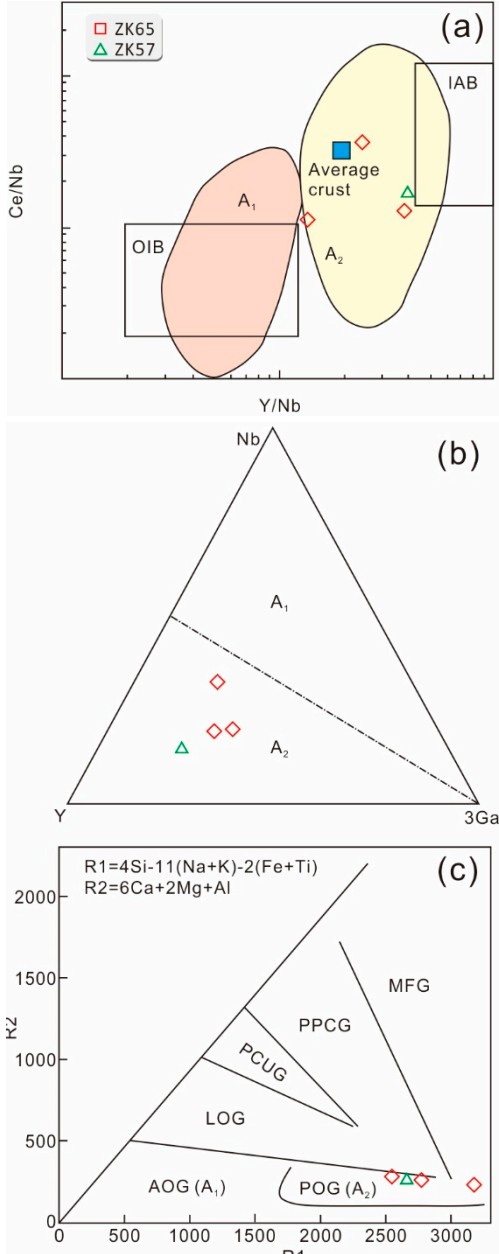

**Figure 11.** (**a**) Plots of Ce/Nb vs. Y/Nb [70]; (**b**) Nb-Y-3Ga [70]; (**c**) the diagrams of Ta-Yb, "R1-R2" [69] for the Xianghualing granites. OIB, ocean island basalt; IAB, island arc basalt; MFG, mantle-fractionated granite; PPG, pre-plate collision granitoids; PUCG, post-collision uplift granitoids; LOG, late-orogenic granitoids; AOG, anorogenic granitoids; POG, post-collision granitoids.

## 5.2. Petrogenesis

The petrogenesis of A-type granites are still debated, and several petrogenetic schemes have been proposed [64,69–75]. The Xianghualing granites have low A/CNK values and no aluminous minerals, which is inconsistent with the metasedimentary-melting petrogenetic model [76]. The rocks exhibit flat heavy REE patterns and high Y contents (116–235 ppm) indicating that garnet was absent from the source reservoir during the partial melting process [73]. The rocks from Xianghualing show weakly peraluminous affinities, which are consistent with crustal magmas produced by partial melting and heated by the underplating of mantle-derived magma [77–79]. This is evident by the obviously different Nd–Hf isotopic compositions between the samples and the coeval mafic rocks (Figures 7a

and 12). Figures 7a and 12 also indicate that the extensive fractional crystallization from coeval mafic magmas for the origin of the A$_2$-type granites can be ruled out.

The granites exclusively display negative zircon $\varepsilon_{Hf}$(t) values, ranging from −6.1 to −2.1, with a corresponding crustal model age of 1.34 to 1.59 Ga (Figure 7b and Table S4). The whole rock Nd model age is 1.47 Ga–1.70 Ga, consistent with the Hf model age. These suggest that the sources of materials had a relatively simple recycling process. The relatively young Nd model ages are a little younger than or similar to that of Mesoproterozoic sediments in South China Block (Figure 7a; T$_{DM}$ = 1.8 Ga) [80]. This indicates a dominated ancient crust material in South China contributed during the magmatic process. The $\varepsilon_{Hf}$(t) vs. $\varepsilon_{Nd}$(t) plots display that the range of granites from Xianghualiang area is similar to the Shi-Hang zone [29–31,33,54,81,82], and the range falls into the field of superimposed areas between global lower crust and global sediments (Figure 12). It suggests the lower crust and/or sediments could be involved during the magmatic process. The lead isotopic values of the four samples from the Xianghualing area display limited variations near the upper crust curve in Figure 13, which is similar to the crust materials in the Shi-Hang zone and the western Pacific subducted sediments at trenches, suggesting that the nature source rocks could be dominated by crustal materials [83]. The high radiogenic lead isotopic compositions in the granites from the Xianghualing area support that the magmas for these granites are predominately derived from crustal materials.

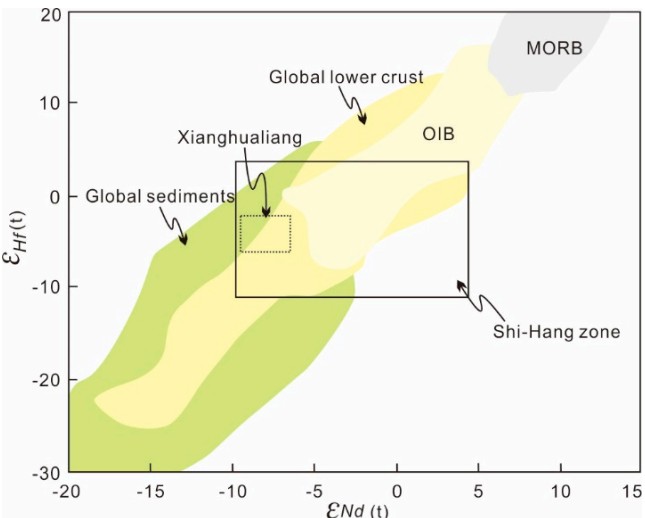

**Figure 12.** $\varepsilon_{Nd}$(t) vs. $\varepsilon_{Hf}$(t) for the granites from Xianghualing area South China. Plots of $\varepsilon_{Nd}$(t) vs. $\varepsilon_{Hf}$(t) fall into the quadrangle. The average values concentrate in the dotted quadrangle. The data of $\varepsilon_{Nd}$(t) and $\varepsilon_{Hf}$(t) are from [29–31,33,54,81,82]. MORB, mid-ocean ridge basalt.

As shown in Table S1 and Figure 11a, the low Gd/Yb ratios and fairly high Y/Nb ratios indicate that the granites formed in the extensional setting [63,84–86]. This is consistent with the support formative processes of A$_2$-type granites, which are considered to crystallize at a high temperature [72,74,75]. The presence of micrographic texture in these A$_2$-type granites is also indicative of a high-level emplacement and provides evidence for an extensional regime (Figure 10) [69,72] and that the magmas formed near the earth surface [68]. A normal geothermal gradient, however, cannot produce high-temperature A-type granites by crustal melting; therefore, an exotic heat source from mantle is a prerequisite. In conclusion, a reasonable explanation for the granites with the signatures of high temperature from the Xianghualing area was generated by partial melting of the crustal materials with minor subducted sediments, and further caused by upwelling of asthenosphere in an extension tectonic setting. The presence of coeval basaltic rocks near the Chenzhou-Lingwu fault supports that a lithospheric extension event could occur during the Early Jurassic Era, such as the Ningyuan alkaline basalts (175 Ma) and the Daoxian basalt (150 Ma) [87–89]. These upwelling mantle materials might provide heat energy for the melting of the crustal materials and subducted sediments.

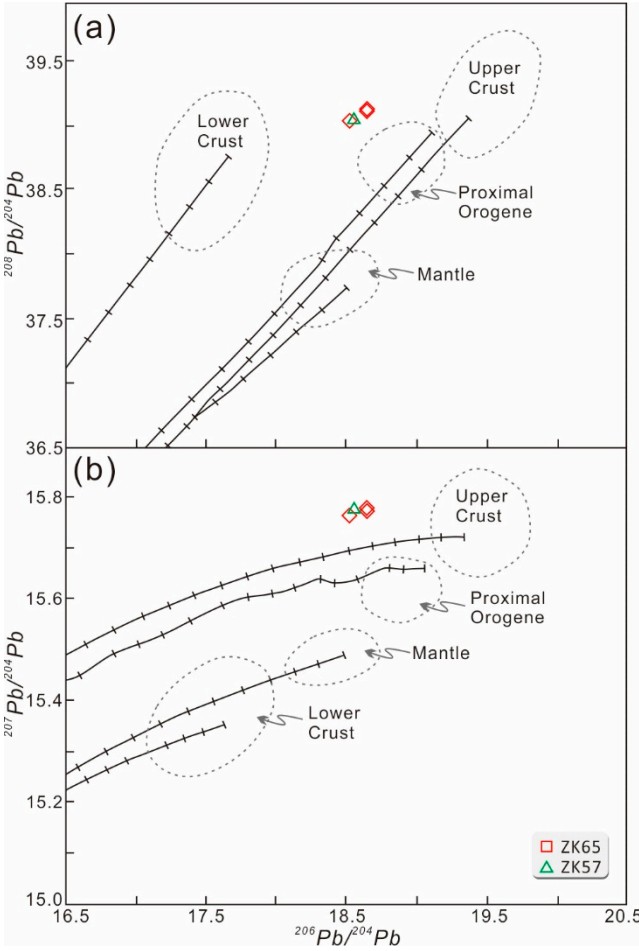

**Figure 13.** Pb isotopic composition of the Jurassic weak peraluminous granites are A-type granites. (**a**) $^{206}$Pb/$^{204}$Pb vs. $^{208}$Pb/$^{204}$Pb diagram of samples; (**b**) $^{206}$Pb/$^{204}$Pb vs. $^{207}$Pb/$^{204}$Pb diagram of samples [83].

*5.3. Tectonic Implications*

The A$_2$-type granite has been generally considered to be derived from the crust in the subduction zone or the continent–continent collision zone [15]. Various models for the formation of A-type granites in South China are proposed in terms of tectonic setting, and these have been correlated with the subduction of the Pale-Pacific Plate for the petrogenesis of the late Mesozoic magma zone [15,24,28,90–94]. Zircon U-Pb dating on granites in the Xianghualing area demonstrate that the granites were emplaced at ~153 Ma. The age is close to those of A-types granites in the Shi-Hang zone, such as Xihuashan (150–153 Ma, [43,44]), Jiufeng (154 ± 1 Ma, [39,40]), Xitian (152 ± 1 Ma, [15]), Qianlishan (152 ± 2 Ma, [18]), Qitianling (155 ± 1 Ma, [36]), Xishan (156 ± 2 Ma, [33]), Jinjiling (156 ± 2 Ma, [19,33]), Huashan (162 ± 2 Ma, [30,31]), and Guposhan (162 ± 2 Ma, [19,32]). These granites formed a north east trending A-type granite belt in the Shi-Hang zone.

We thus propose that these A$_2$-type granites could be formed in an extensional setting (Figure 1d). The Paleo-Pacific Plate subducted underneath the SE China Block at a very low angle, beginning in the Late Triassic Era and reaching southern Hunan at ca. 174 Ma (Figure 14a) [24,94,95]. The low angle subduction could have caused crustal thickening in the coastal area. Due to the temperature and pressure increases, the ferron (A-type) granitoids were emplaced in the orogenic thermal relaxation regime [24]. From ca. 174 to 164 Ma, the slab dip angle increased and subsequently caused the subducting slab to dehydrate and release fluids. The slab-released fluids triggered partial melting of mantle wedge and generated basaltic magmas. Around or above the melt zone, the Middle

Jurassic I- and S-type magmas formed in the region, such as in Tongshanling (~164 Ma) in Southern Hunan (Figures 1d and 14b) [19]. The rollback of the Paleo-Pacific plate led to a regional extension during the late Jurassic Era (163–150 Ma) (Figure 14c) [18,19,61,92,93,96]. The extension caused the crust and lithospheric mantle to become thinner, with an accompanying asthenosphere upwelling. The upwelling of the basaltic magmas might have provided heat energy and triggered a partial melting of the thinned crust rocks and subducted sediments to form the ca. 163–150 Ma $A_2$-type granites along the Shi-Hang zone.

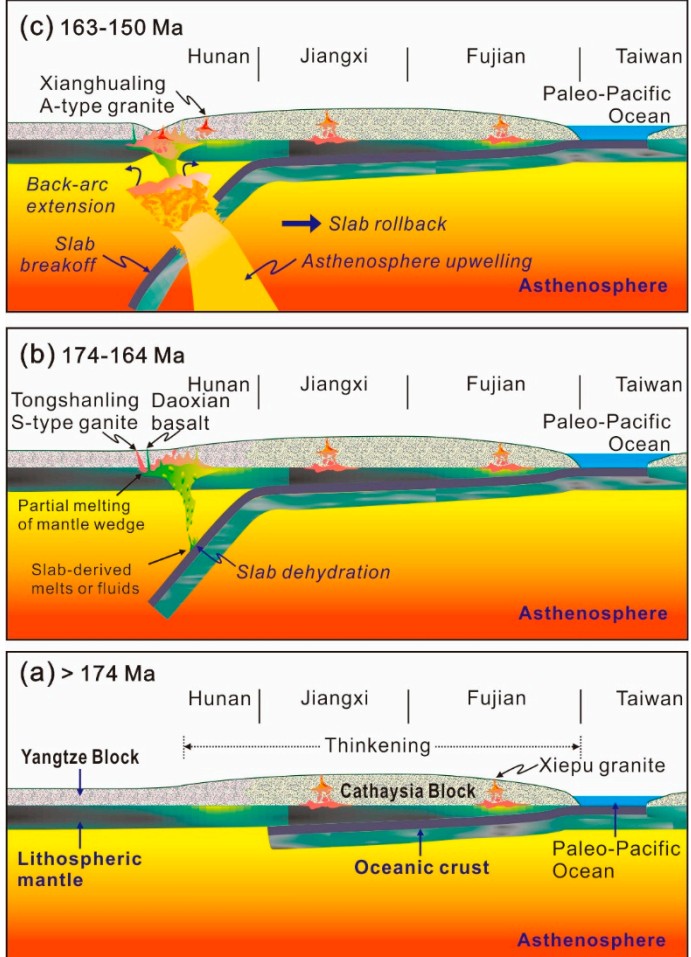

**Figure 14.** Schematic illustrations showing the generation of the late Jurassic A-type granites in South China. (**a**) Low angle subduction before ~174 Ma, forming the middle Jurassic S-type granites such as the Xiepu granite; (**b**) High-angle subduction during 174 to 164 Ma, forming S-type granites and basalt such as the Tongshanling S-type granite and the Daoxian basalt; (**c**) During the late Jurassic Era, slab rollback and slab breakoff formed the A-type granite such as the Xianghualing granite.

*5.4. Implication for the Generation of Tin Mineralization*

Yuan et al. [12] gave a cassiterite U–Pb weighted average age of 156 ± 4 Ma. The ore-forming age coincidesdwith the other W–Sn–Pb–Zn deposits in the Shi-Hang zone, such as the Shizhuyuan W deposit (Re–Os, 151 ± 4 Ma, [97]; Sm–Nd, 161 ± 2 Ma, [98]; Sm–Nd, 149 ± 2 Ma, [41]), the Furong tin ore field ($^{40}$Ar–$^{39}$Ar, 160–150 Ma, [50]), the Yaoganxian tungsten deposit (Re–Os, 150 ± 3 Ma, [99]), the Huangshaping Pb–Zn–W–Sn–Ag deposit (Re–Os, 155 ± 2 Ma, [38]), the Yuanzhuding Cu–Mo deposit (Re–Os, 157 ± 4 Ma, [47]), and the Xitian W–Sn deposit ($^{40}$Ar–$^{39}$Ar, 157–150 Ma and Cassiterite U–Pb, 154–157 Ma [100–102]). These ages indicate that a globally significant W-Sn polymetallic mineralization has a close relationship with the A-type granites in the Shi-Hang zone. It is further

supported by the fact that the Sn, W, Pb, and Zn contents of the Mesozoic granites are more than ten times their Clark values, respectively. For examples, the Sn, W, Pb, and Zn contents of the A-type granites in the study area are 8.7–39.7 ppm, 34.4–66.7 ppm, 48.2–66.1 ppm, and 15.0–44.9 ppm, respectively (Table S1). It suggests that they could provide ore-forming materials for mineralization. Moreover, the later Jurassic A-type granites are characterized by high Si (74.1 wt.% to 78.0 wt.%) and Rb (1606 ppm to 2351 ppm) contents, low Ti, P, Sr, and Ba contents, extremely high Rb/Sr ration and flat heavy REE (HREE) with strong negative Eu anomalies. These features imply that an intense fluid–magma interaction is favorable to the formation of the Sn–W-polymetallic mineralization [103, 104]. In the Xianghualing deposit, skarn rocks containing fluorite markedly developed around the granite porphyry, indicating the involvement of an F-rich fluid. The dramatic negative Eu anomaly may indicate that the ore-forming fluids were developed in oxidizing conditions [12]. All of these indicate that there is huge W–Sn prospecting potential related to the granites in the Xianghualing deposit.

## 6. Conclusions

(1) The low Sr and high Yb $A_2$-type granites from the Xianghualing Sn polymetallic deposit were emplaced in the later Jurassic Era (~153 Ma) with a typical enrichment in $SiO_2$, REEs (except Eu), HFSE, and Yb, and a depletion in $Al_2O_3$, CaO, MgO, and Sr.

(2) The geochemical data and isotopic composition (Sr–Nd–Pb–Hf) suggest that the granites from the Xianghualing deposit were derived from predominantly crustal materials (Mesoproterozoic basement rocks in South China Block and subducted sediments) and some minor subducted sediments.

(3) Crustal partial melting in the extensional tectonic setting was induced by subduction of the Paleo-Pacific Plate, accompanied by the decompression melting in the localized mantle wedge and a rollback of the Paleo-Pacific Plate. These would be the likely accepted combined mechanisms for the petrogenesis of $A_2$-type granites.

**Supplementary Materials:** The following are available online at http://www.mdpi.com/2075-163X/9/3/182/s1, Table S1: Whole-rock major and trace element of the low-Sr and high-Yb granites in the Xianghualing, SW China; Table S2: Sr, Nd and Pb isotope data for selected granites from Xiahualing, South China. Table S3: LA–ICP–MS zircon U–Pb age data of the late Jurassic low-Sr and high-Yb A-type granites from Xianghualing, South China; Table S4: In situ zircon Hf isotope data for the later Jurassic low-Sr and high-Yb A-type granites in the Xianghualing Sn-polymetallic deposit, South China.

**Author Contributions:** Conceptualization, C.H.X. and C.S.W.; Data curation, C.H.X. and Y.K.S.; Investigation, C.H.X. and C.S.W.; Project administration, Y.K.S.; Writing—original draft, C.H.X.; Writing—review & editing, C.H.X.

**Funding:** This research was jointly funded by the National Key R&D Program of China (No. 2016YFC0600107) and the Basic Research Fund of the Chinese Academy of Geological Sciences (No. JYYWF20180602, JYYWF20183702).

**Acknowledgments:** We would like to thank Qing Zhang, Kun-Feng Qiu, and two anonymous reviewers for the constructive comments and suggestions on this paper.

**Conflicts of Interest:** The authors declare no conflict of interest.

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
