# Peer review of "Petrogenesis of Low Sr and High Yb A-Type Granitoids in the Xianghualing Sn Polymetallic Deposit, South China: Constrains from Geochronology and Sr–Nd–Pb–Hf Isotopes"

_minerals, doi:10.3390/min9030182_

Round 1
Reviewer 1 Report
General comments
1. Do not try to overstate the precision in the text. For example in the abstract:
Text: Zircon U–Pb dating indicates that these granites emplaced at 152.7~152.8 Ma”
Comment: LA-ICP-MS is not able to provide such precision and, especially, accuracy. Write ~ 153 Ma.
2. Text: 11.04 wt.%–12.66 wt.%;
Comment: Here and everywhere in the text for major elements, use only one significant number after the decimal point (11.0-12.7 wt.%). The second number is an overestimation of the precision. The same is true for the trace elements (for example, write 154-6771 ppm instead of 154.29-6771.43 ppm in line 186 and everywhere when similar data are shown).
3. Use only initial isotopic values in the discussion. Keep measured values for the data tables only.
4. Lines 156-158: zircon saturation thermometer “is a theoretical temperature and cannot be used to constrain magmatic or partial melting temperatures” (Siegel et al. 2018 Earth-Science Reviews, v. 176, p. 87-116). The same applies to lines 247 and 286.
5. Lines 266-271: If you assume mixing of mantle and crustal lithologies in the generation of the studied granite magma, you cannot directly compare the TDM model ages with age of the crust, because you model ages will always be younger depending on the amount of mantle component in the mixture.
6. In section 5.3 you discuss the extensional tectonic setting for Shi-Hang zone created in relation to the rollback of the Paleo-Pacific plate. However, the extension at ca 150 Ma has already started in other regions of East and Central Asia (see Ivanov et al., 2015 Earth-Sci Rev, v. 148, p. 18-43; Ivanov et al., 2018 Minerals, v. 8, 602). Could it have a general cause, such as the collapse of the Mongol-Okhotsk orogeny and reorganization of microplates in Asia?
Particular comment:
Abstract: exhibit extremely high 87Sr/86Sr ratio (1.13–1.89)
Comment: That is obviously uncorrected value to the radiogenic accumulation of 87Sr. Remove it from the abstract.
Line 147: dEu should be defined how it was calculated.
Line 148: DI should be defined how it was calculated.
Lines 163-166: The problem that you cannot obtain realistic initial Sr isotopic ratios is not only due to high Rb/Sr ratio but also due to post-eruption modification of this ratio. Negative initial values are senseless.
There is no section 4.1.
Section 4.2 “U-Pb age” should precede the section on geochemistry.
Line 191: 105.37-156.35? Should it be ~155-156?
Lines 222-223: Of course there is no relation between P and Si because your rocks contain almost no P.
English is understandable but requires extensive language editing. Below are some suggestions for Introduction, but the text should be corrected overall. There are numerous typos and grammar mistakes.
Abstract: These features indicate that the low–Sr and high–Yb weakly peraluminous A2–type granites were generated by the overlying partial melting in the extensional tectonic setting,
Suggest re-writing “We interpret these features that the low–Sr and high–Yb weakly peraluminous A2–type granites were generated by the overlying partial melting in the extensional tectonic setting, …
Line 36: The Mesozoic era window
Delete ‘era window’
Line 39: low Nd model ages
Low? Young/old? Provide the values.
Line 40: less negative
Delete ‘less’
Line 43: on the earth
Change to ‘on Earth’
Lines 43-44: spatially associated [what? Intrusions?]
Line 45: suggest their variations in the petrogenesis and tectonic regimes
Change to - suggest different petrogenesis and tectonic regimes
Line 63: Late–Jurassic granites related
Change to Late–Jurassic granite-related
Line 66: evidence instead of evidences
Author Response
Replies to Reviewer
The valuable revisions and constructive comments from you are highly appreciated. We have incorporated all the comments and suggestions and have revised the manuscript earnestly and carefully.
General comments
1. Do not try to overstate the precision in the text. For example in the abstract:
Text: Zircon U–Pb dating indicates that these granites emplaced at 152.7~152.8 Ma”
Comment: LA-ICP-MS is not able to provide such precision and, especially, accuracy. Write ~ 153 Ma.
Reply: This constructive suggestion is helpful for us. We have revised the precision in the text. In the updated manuscript, we use the ages in the abstract and discussion with no significant number.
2. Text: 11.04 wt.%–12.66 wt.%;
Comment: Here and everywhere in the text for major elements, use only one significant number after the decimal point (11.0-12.7 wt.%). The second number is an overestimation of the precision. The same is true for the trace elements (for example, write 154-6771 ppm instead of 154.29-6771.43 ppm in line 186 and everywhere when similar data are shown).
Reply: This constructive suggestion is deeply appreciated. In the updated manuscript, we modified all the data expressing mode of major elements and trace elements.
3. Use only initial isotopic values in the discussion. Keep measured values for the data tables only.
Reply: This constructive suggestion is helpful for us. We use only initial Sr isotopic values in the discussion and remove the 87Rb/86Sr and (87Sr/86Sr)i data in the Table S2.
4. Lines 156-158: zircon saturation thermometer “is a theoretical temperature and cannot be used to constrain magmatic or partial melting temperatures” (Siegel et al. 2018 Earth-Science Reviews, v. 176, p. 87-116). The same applies to lines 247 and 286.
Reply: This constructive suggestion is helpful for us and is deeply appreciated. We remove all of the zircon saturation temperatures in this paper.
5. Lines 266-271: If you assume mixing of mantle and crustal lithologies in the generation of the studied granite magma, you cannot directly compare the TDM model ages with age of the crust, because you model ages will always be younger depending on the amount of mantle component in the mixture.
Reply: This constructive suggestion is deeply appreciated. In the updated manuscript, we agree with your opinion and revise the model of the petrogenesisi of the granites in the Xianghualiang area. We emphasize the upwelling mantle materials might provide heat energy for the melting of the crustal materials and subducted sediments. The origin of the granites is major derived mainly from crustal materials.
6. In section 5.3 you discuss the extensional tectonic setting for Shi-Hang zone created in relation to the rollback of the Paleo-Pacific plate. However, the extension at ca 150 Ma has already started in other regions of East and Central Asia (see Ivanov et al., 2015 Earth-Sci Rev, v. 148, p. 18-43; Ivanov et al., 2018 Minerals, v. 8, 602). Could it have a general cause, such as the collapse of the Mongol-Okhotsk orogeny and reorganization of microplates in Asia?
Reply: Thanks for your advice. We learn the two papers you recommended. In the paper (Ivanov et al., 2018 Minerals, v. 8, 602) pointed that the Early Cretaceous alkaline magmatism were widespread within the Aldan Shield of the Siberian Craton and its surrounding along the Mongolia–Okhotsk suture zone under the post-collisional extension. The age of these intrusions range from 150 to 120 Ma which is younger than the intrusions in South China. These intrusions may form in the same tectonic background with the Later Cretaceous intrusions in South China. Owing to the knowledge limitations, we do not rule out the possibility that the collapse of the Mongol-Okhotsk orogeny caused the extensional tectonic setting for Shi-Hang zone.
Particular comment:
1. Abstract: exhibit extremely high 87Sr/86Sr ratio (1.13–1.89)
Comment: That is obviously uncorrected value to the radiogenic accumulation of 87Sr. Remove it from the abstract.
Reply: We remove it from the abstract.
2. Line 147: δEu should be defined how it was calculated.
Line 148: DI should be defined how it was calculated.
Reply: The calculation formula of δEu and DI were provided in the Table S1.
3. Lines 163-166: The problem that you cannot obtain realistic initial Sr isotopic ratios is not only due to high Rb/Sr ratio but also due to post-eruption modification of this ratio. Negative initial values are senseless.
Reply: This constructive suggestion is deeply appreciated. We use only initial Sr isotopic values in the discussion and remove the 87Rb/86Sr and (87Sr/86Sr)i data in the Table S2.
4. There is no section 4.1.
Section 4.2 “U-Pb age” should precede the section on geochemistry.
Reply: We apologize for the carelessness in the order of the section. In this paper, the Whole-rock geochemistry is section 4.1.
5. Line 191: 105.37-156.35? Should it be ~155-156?
Reply: line 191 “105.37-156.35” represent the single 206Pb/238U age of ZK65 which display in the Table S3.
6. Lines 222-223: Of course there is no relation between P and Si because your rocks contain almost no P.
Reply: Thanks for your comment. We remove this sentence in this discussion.
7. English is understandable but requires extensive language editing. Below are some suggestions for Introduction, but the text should be corrected overall. There are numerous typos and grammar mistakes.
Reply: We improved the language one by one in the helping of my colleague who studied and worked in the United States. We try our best to improve the language.
8. Abstract: These features indicate that the low–Sr and high–Yb weakly peraluminous A2–type granites were generated by the overlying partial melting in the extensional tectonic setting,
Suggest re-writing “We interpret these features that the low–Sr and high–Yb weakly peraluminous A2–type granites were generated by the overlying partial melting in the extensional tectonic setting, …
Reply: This constructive suggestion is deeply appreciated and accepts this suggestion.
9. Line 36: The Mesozoic era window
Delete ‘era window’
10. Line 39: low Nd model ages
Low? Young/old? Provide the values.
11. Line 40: less negative
Delete ‘less’
12. Line 43: on the earth
Change to ‘on Earth’
Reply: We accept and revise the suggestion of 9 to 12.
13. Lines 43-44: spatially associated [what? Intrusions?]
Reply: We change the sentence to “Previous studies defined that these temporally and spatially associated intrusions are different in terms of petrography, and elemental and isotopic compositions, and argued for different petrogenesis and tectonic regimes, ranging from an extension, rift–related and convergent, subduction–related models (Figure 1d), to collapse of orogeny models.”.
14. Line 45: suggest their variations in the petrogenesis and tectonic regimes
Change to - suggest different petrogenesis and tectonic regimes
Reply: We accept and revise the suggestion.
15. Line 63: Late–Jurassic granites related
Change to Late–Jurassic granite-related
Reply: We accept and revise the suggestion.
16. Line 66: evidence instead of evidences
Reply: We accept and revise the suggestion.

Reviewer 2 Report
Comments on the manuscript “Petrogenesis of low–Sr and high–Yb A–type granitoids in the Xianghualing Sn-polymetallic deposit, South China: constrains from geochronology and Sr-Nd-Pb-Hf isotopes”, by Chang-Hao Xiao, Yu-Ke Shen, and Chang-Shan Wei.
This research paper provides interesting data on the petrogenesis of A2-type granites related to Sn-W-polymetallic mineralization of the Jurassic widespread Yanshanian granitoids of the South China Block.
Observations and suggestions are detailed as a commented .pdf attached to this report.
The authors built up their work on four granite samples extracted from two drill cores of about 200 to 300 m depth. The applied methodology to study the samples is appropriate and all details regarding the analytical techniques were provided in supplementary tables.
Their most remarkable and novel contribution would be its unusual whole rock trace element chemistry, specifically its low Sr and high Yb contents. The other trace element and major whole rock geochemistry fits well within “normal” or expected compositions for other worldwide A-type granites emplaced in an extensional regime. To my opinion, the strongest new data the authors contribute is their whole rock Sr-Nd-Pb and zircon U-Pb-Hf isotope geochemistry, which allowed them to arrive to a quite good and accepted interpretation and conclusion regarding the combination of processes that best explain the origin of the melt and the geotectonic emplacement environment. I have to say that I am not familiar with this region of Asia, except for being aware of the important granite-related W-Sn-mineralizations. For this reason I want to make it clear that I think that the authors´ conclusions are well-supported with data of good quality, but I do not have arguments strong enough to be discussed with other colleagues that have worked in that region about the details of the author´s interpreted data, in particular the tectonic implications.
The authors have made a detailed comprehensive review of previous work in the region, and that is shown in an extensive list of cited updated bibliography; general bibliography related to A-type granites was referenced as well.
The relationship between the A2-granites and Sn-W mineralization is well established. The ca. 153 Ma U-Pb age of zircons does not pose any surprise because it fits well within a previously well known set of ages for other Jurassic neighboring intrusives, apparently of the same magmatic cycle.
The only thing about this paper that might deserve some criticism is that this contribution lacks a section devoted to traditional petrography and, specially, to accessory mineral identification. The authors evidence at the end of their manuscript (Discussion, 5.1, lines 229-231), that a couple of mineralogical-petrographic features were essential, or at least important, to support the A nature of these granites.
Furthermore, the REE geochemistry has to be to some extent controlled by accessory minerals. Except for zircon, which has been devoted a special section as a trace element key indicator of age and genetic processes, any other accessory minerals have been mentioned or described; for instance, I wonder about monazite presence in these rocks and its implications or meaning in the discriminatory graphs of Whalen et al. (1987) and Eby (1992). With respect to the essential mineralogy of these granites, the data provided is elementary (Sample description, 3.1, lines 101-109). I would have expected to see here, instead, a section devoted to Petrography, most likely with some mineral chemistry data at least addressed to the composition of feldspar, biotite and Ca-Amphibole. Even though I understand that this paper is mainly based on whole rock major-, trace-, and isotope- geochemistry, as it is clearly stated in its title, any petrogenetic work should have a solid petrographic/mineralogical basis that could help to better understand the whole rock geochemistry.

Author Response
Replies to Reviewer
The valuable revisions and constructive comments from you are highly appreciated. We have incorporated all the comments and suggestions and have revised the manuscript earnestly and carefully.
1. The only thing about this paper that might deserve some criticism is that this contribution lacks a section devoted to traditional petrography and, specially, to accessory mineral identification. The authors evidence at the end of their manuscript (Discussion, 5.1, lines 229-231), that a couple of mineralogical-petrographic features were essential, or at least important, to support the A nature of these granites.
Furthermore, the REE geochemistry has to be to some extent controlled by accessory minerals. Except for zircon, which has been devoted a special section as a trace element key indicator of age and genetic processes, any other accessory minerals have been mentioned or described; for instance, I wonder about monazite presence in these rocks and its implications or meaning in the discriminatory graphs of Whalen et al. (1987) and Eby (1992). With respect to the essential mineralogy of these granites, the data provided is elementary (Sample description, 3.1, lines 101-109). I would have expected to see here, instead, a section devoted to Petrography, most likely with some mineral chemistry data at least addressed to the composition of feldspar, biotite and Ca-Amphibole. Even though I understand that this paper is mainly based on whole rock major-, trace-, and isotope- geochemistry, as it is clearly stated in its title, any petrogenetic work should have a solid petrographic/mineralogical basis that could help to better understand the whole rock geochemistry.
Reply: This constructive suggestion is helpful for us. We agree with your opinion. We supple micrograph observation. Accessory minerals are composed of zircon with minor apatite. We do not find monazite in these rocks. In this paper, mineral chemistry to determine the composition of feldspar, biotite and Ca-Amphibole is an important work to support the A nature of these granites. But we do not do this work before. However, the micrographic texture and whole rock major-, trace-, and isotope- geochemistry could support our point. This constructive suggestion will be used in the future similar works.
2. line 47-60
Reply: We revise the Figure 1 according to the suggestion of Reviewer. And insert label (b) and (c).
3. line 78-81
Reply: We revise the sentence according to the suggestion of Reviewer.
4. line 84-93
Reply: we want to introduce the location and geological feature of the Xianghualing Sn-polymetallic deposit in this paragraph. The structural, stratigraphic, and magmatic features were observed by us. So we do not add references in this paragraph.
We relocate the sentence “The Xinfeng mine is one of the most important mines in the Xianghualing deposit” after first sentence on this paragraph.
5. line 96-106
Reply: This is a typos mistake about the texture of the granites which are porphyritic texture.
6. Reply: We revised all of the Figures and remove the Figure 12 according to the suggestion of Reviewer in the updated manuscript.
7. line 243-244 “You should discuss the meaning of these textures”
Reply: The classic interpretation on the development of myrmekite is stress concentration. Myrmkite tubules develop in response to applied stresses, forming a corona around the edge of the feldspar, mostly plagioclase (Simpson, 1985; Simpson and Wintsch, 1989). Myrmkite and myrmekite corona are commonly observed in mylonites and contact metamorphism (Simpson and Wintsch, 1989; Vernon, 1991; Cesare et al., 2002). If the processes is complete, feldspar grain can be totally replaced by myrmekite. The occurrence of myrmekite is indicative of high temperature but unnecessarily decompression. We just display that this phenomenon is common and we do not discuss the meaning of these textures.
9. Reply: typos and grammar mistakes pointed in the paper have revised in the updated manuscript with marked.

Round 2
Reviewer 1 Report
The authors have improved their m/s significantly. However, I still have some comments.
I appreciate that the authors referred to two of our publications in introduction (lines 47-53), however the way it is done is inapropriate. We did not study granites of SE China, thus the references to our publications should be either removed or the text modified. The reason that I provided references to these publication was to show that the extentional regime at the same period of time was not only in SE China, which is the focus of the reviewed m/s, and also elswhere in Central and East Asia along previous plate margins. So, the extention could have large regional significance and thus the driving force(s) could be more than just the subduction from east.
In supplementary material I did not understand the following sentence: The analytical precisions were ~1–3% for elements at N1.0 wt.% concentration and ~10% for elements b1.0 wt.%.
What is N and b here? Should it be like N.0 wt.% and 0.N wt.%?
Author Response
Dear Reviewer:
The valuable revisions and constructive comments from you are highly appreciated. We have incorporated the two comments and suggestions and have revised the manuscript earnestly and carefully.
Point 1: I appreciate that the authors referred to two of our publications in introduction (lines 47-53), however the way it is done is inapropriate. We did not study granites of SE China, thus the references to our publications should be either removed or the text modified. The reason that I provided references to these publications was to show that the extentional regime at the same period of time was not only in SE China, which is the focus of the reviewed m/s, and also elswhere in Central and East Asia along previous plate margins. So, the extention could have large regional significance and thus the driving force(s) could be more than just the subduction from east.
Response 1: The constructive suggestion is deeply appreciated. Initially, we cited two of the Reviewer’s publications because the author pointed the Early Cretaceous alkaline magmatism were widespread within the Aldan Shield of the Siberian Craton and its surrounding along the Mongolia–Okhotsk suture zone under the post-collisional extension. The age of these intrusions ranges from 150 to 120 Ma which is younger than the intrusions in South China. We think the tectonic background is not similar with the alkaline magmatism in the Reviewer’s publications. But we have to say that we are not familiar with this region of Asia, except for China. We do not rule out the possibility that the collapse of the Mongol-Okhotsk orogeny caused the extensional tectonic setting for Shi-Hang zone. This is the reason we referred to two of Reviewer’s publications. If we misunderstanding and unsuitable to cite the references, we apologize to the authors and removed the references in our paper.
Point 2: In supplementary material I did not understand the following sentence: The analytical precisions were ~1–3% for elements at N1.0 wt.% concentration and ~10% for elements b1.0 wt.%.
What is N and b here? Should it be like N.0 wt.% and 0.N wt.%?
Response 2: This constructive suggestion is deeply appreciated. We apologize for the typos. It may cause by the character display error. The sentence changes to “The analytical precisions were ~1–3% for elements at >10 wt% concentration and ~10% for elements <1.0 wt%.”